# The LRINEC Score—An Indicator for the Course and Prognosis of Necrotizing Fasciitis?

**DOI:** 10.3390/jcm11133583

**Published:** 2022-06-22

**Authors:** Vanessa Hoesl, Sally Kempa, Lukas Prantl, Kathrin Ochsenbauer, Julian Hoesl, Andreas Kehrer, Talia Bosselmann

**Affiliations:** 1Center of Plastic, Hand and Reconstructive Surgery, University Hospital Regensburg, 93053 Regensburg, Germany; sally.kempa@ukr.de (S.K.); lukas.prantl@ukr.de (L.P.); kathrin.ochsenbauer@yahoo.de (K.O.); talia.bosselmann@ukr.de (T.B.); 2Faculty of Medicine, University of Regensburg, 93053 Regensburg, Germany; julian_hoesl@yahoo.de; 3Section of Plastic Surgery, Hospital Ingolstadt, 85049 Ingolstadt, Germany; andreaskehrer@gmx.de

**Keywords:** necrotizing fasciitis, soft tissue infection, LRINEC, debridement

## Abstract

Background: The Laboratory Risk Indicator for Necrotizing Fasciitis score (LRINEC) is a simple tool used to support early diagnosis of Necrotizing Fasciitis (NF). The aim of this study was to investigate whether the LRINEC is suitable as a progression and prognosis parameter in patients with NF. Methods: In this retrospective study, laboratory data of 70 patients with NF were analyzed. The LRINEC was calculated for every patient at the time of hospital admission and postoperatively after surgical interventions. Furthermore, the LRINEC was examined as a prognostic factor for survival. Results: The overall lethality of our series was 20 out of 70 (28.6%). A highly significant LRINEC decrease was found for serial debridements. The largest decrease was observed after the first debridement. There was a significant difference between the initial LRINEC of deceased and surviving patients. A cut off value of >6.5 (7 LRINEC points) resulted in an optimal constellation of sensitivity (70%) and specificity (60%) to predict lethality in patients with NF. Conclusions: The LRINEC significantly decreases after surgical debridement. An initial LRINEC equal or greater than seven is an independent prognostic marker for lethality and can help to identify high-risk patients.

## 1. Introduction

Necrotizing Fasciitis (NF) is a rare but fulminant, life-threatening soft tissue infection, that is characterized by rapidly progressing necrosis of subcutaneous tissue and the deep layers of fascia, resulting in severe systemic infection [1]. NF is rare, with an incidence of four cases per 100,000 population. In contrast to this, the lethality rate ranges from 20 to 30% and may reach up to 100% if there are delays in diagnosis and treatment [2,3,4,5]. The infection is commonly located in the lower extremities, the trunk, and the perineum [6,7,8] and caused by aerobic and anaerobic bacteria, group A β-hemolytic streptococci, vibrios, aeromonas and fungal germs. Comorbidities associated with NF are, for example, diabetes mellitus, advanced age, atherosclerosis, liver cirrhosis, heart failure, chronic kidney failure, obesity, and immunosuppression [2].

To prevent rapid progression of the infection, and thus improve the patients’ prognosis, the disease must be diagnosed at an early stage and immediate surgical treatment must be initiated [9]. Diagnosis is primarily based on physical examination and is supplemented by laboratory or computed tomography findings [10]. The clinical appearance of necrotizing fasciitis is, however, underestimated in the initial stages of the disease due to a lack of specific clinical features and characteristics, and it is mistaken for other skin and subcutaneous infections, such as cellulitis or erysipelas [11]. Main clinical symptoms consist of local symptoms, such as erythema, swelling, blistering with serous fluid, tissue hardening, bluish dark skin discoloration, hemorrhagic bullae, and skin necrosis. Furthermore, initial stages of necrotizing fasciitis are also characterized by an exquisitely severe pain that is out of proportion to the results of the physical examination [11]. In the course of the disease, virulent organisms and toxins are released from the infected tissue into the bloodstream, triggering a systemic toxic reaction [11].

The combination of aggressive surgical debridement, broad systemic antibiotic therapy, and a multidisciplinary approach of intensive care with active fluid substitution and control of sepsis is decisive for successful therapy [3,12,13]. Since not all patients recover with a single debridement, it is advisable to continue reevaluations at intervals of six to 48 h until no further necrosis is visible, and all infected tissue has been removed [13]. Furthermore, precise monitoring of patient physiology and serial white blood cell count should be carried out every six to twelve hours, as patients often develop organ failure, for which replacement therapy is necessary [13]. Since the timely clinical diagnosis of necrotizing fasciitis proves to be extremely difficult and delayed detection is one of the main reasons for high lethality rate of these soft tissue infections, in 2004, Wong et al. described a diagnostic evaluation system, named “Laboratory Risk Indicator for Necrotizing Fasciitis” (LRINEC) score [14]. It is based on six laboratory tests (Table 1) and stratifies patients into low, medium, or high risk of NF (Table 2) [14]. The score should support early diagnosis and differentiation from other severe soft tissue infections that require differentiated treatment [14]. The initial study showed excellent positive and negative predictive value, but in recent years the validity of the LRINEC score and its role as a scoring system were the subject of repeated controversy in numerous studies [14,15].

The aim of this study was the descriptive characterization of patients who underwent surgical treatment due to a histopathologically confirmed diagnosis of necrotizing fasciitis. In addition to the initial values of the LRINEC at the time of hospital admission, the course of the LRINEC score, postoperatively after multiple surgical interventions, is presented for the first time in this study. Our aim was to demonstrate that the already validated positive effect of surgical therapy in patients with necrotizing fasciitis is evident from the changes in the LRINEC score in the course of time. In a further step, we investigated to what extent the LRINEC (course) correlates with the clinical course of the patients, or is influenced by certain factors and patient characteristics, to be able to assess the LRINEC score in its potential role as a progression and forecast parameter.

## 2. Materials and Methods

### 2.1. Patient Selection

For this study, data were collected retrospectively from all patients that were hospitalized and surgically treated between January 2009 and December 2019 at Level-I-trauma center, University Medical Center Regensburg, Germany, due to a confirmed diagnosis of necrotizing fasciitis. The clinical diagnosis was confirmed by microbiologic germ detection. Standard treatment after admission included surgical debridement, as well as broad-spectrum antibiotics based on microbiological findings. Critical care support was required for most of the patients.

After collection of all relevant parameters, a total of 70 patients could be included in this study. A total of 103 patients had to be excluded due to missing data, inconclusive diagnosis, or initial surgical treatments performed at referring hospitals. Each patient’s record was analyzed for the following: age at admission, gender, presence of predisposing factors and comorbidities, anatomic site of infection, microbiological findings, histopathological findings, duration of hospitalization, intensive care unit stay, in-hospital lethality rate, frequency and type of operative procedures performed, time from admission to operative treatment, need for amputation, type of antibiotic therapy, number of hyperbaric oxygen therapy (HBO) and vacuum-assisted-closure therapy (VAC) administered, as well as laboratory results for calculation of the LRINEC.

### 2.2. Calculation of the LRINEC

The LRINEC was calculated retrospectively for each patient, depending on the availability of laboratory findings at the time of admission, and postoperatively after the first three debridements (Table 3).

### 2.3. Statistical Methods

Full statistical analysis was performed using Microsoft Excel (Microsoft Excel for Mac, version 16.50, Redmond, WA, USA) and SPSS (IBM^®^ SPSS^®^ Statistics for Macintosh, version 25.0, Armonk, NY, USA). Patient characteristics were presented using absolute and relative frequencies, arithmetic mean, standard deviation, median, and range. Descriptive representations of the LRINEC were made for metric variables using tables of distribution characteristics (mean, median, standard deviation, minimum, and maximum) and for categorical variables as cross-tabulations and using charts. Boxplots and line plots were specifically used to visualize the LRINEC score over time. To visualize the LRINEC time course, longitudinal data were tested using multilevel models. Linear regressions were used to examine the impact of metric influencing factors (for example, days between inpatient admission and first surgery) on the change in LRINEC score. Mean comparisons between patient groups were made using the *t*-test. To test the normal distribution of the variables, a Q–Q plot was created in each case. To examine the initial LRINEC score as a prognostic factor for survival, ROC analysis was performed. Furthermore, a Kaplan–Meier evaluation was used to reveal differences in survival between patients with high and patients with low LRINEC scores on admission. Survival in these two patient groups was plotted using Kaplan–Meier curves and differences were tested using log-rank tests or Tarone–Ware tests (if the K–M curves crossed). A *p*-value of ≤0.05 was considered statistically significant.

The study collective decreased due to death but mainly due to exclusion because of a lack of data. Of the 70 patients, 20 (28.5%) died during their hospital stay. A total of 36 patients had complete data of their LRINEC score at any time. The partial exclusion of the other 34 patients was taken into account according to the Kaplan–Meier estimate. Because of the retrospective design of the study, exclusion was due to the lack of laboratory parameters. This was the main reason for using the Kaplan–Meier estimation. In this way, the resulting bias due to the omission of the data could be avoided as much as possible.

## 3. Results

### 3.1. Characteristics of the Study Group

A total of 70 patients diagnosed with necrotizing fasciitis were enrolled and analyzed. Of these, 43 were male patients (61.4%) and 27 were female patients (38.6%). The mean age was 58.4 ± 15.9 years (range 8–89 years) at the time of hospital admission. The overall lethality rate of our series was 28.6% (n = 20). Of these, two patients died within the first 24 h, four patients died after a hospital stay of more than 80 days. Surviving patients were discharged home or transferred to a rehabilitation facility after a mean hospital stay of 49.9 days ± 41.6 (median 37 days, range 9–230). A stay in the intensive care unit (ICU) was recorded for 47 patients (67.1%). Seven patients (14.9%) received intensive care only 24 h postoperatively after surgical debridement. The most common comorbidity was arterial hypertension in 33 patients (47.1%), while diabetes mellitus was present in 18 patients (25.7%). In addition, 17 patients (24.3%) suffered from obesity and 16 patients (22.9%) from chronic kidney disease. The remaining 10 patients (14.3%) did not show any predisposing factors. The most common anatomical site of infection in the entire cohort were the lower extremities, which compromised 62.9%, followed by the trunk (42.9%) and upper extremities (21.4%). In only one patient, were the head and neck affected.

All 70 patients had positive microbiological cultures. We found polymicrobial organisms in 50 patients (71.4%), the most common pathogen being staphylococcus aureus in 20 cases (28.6%). Surgical interventions were carried out in all patients. The mean number of debridement procedures needed was 5.4 ± 3.1 (range 1–15). The mean time from admission to first debridement was 1.2 ± 1.9 days. A total of 36 patients (51.4%) had surgery on the day of admission, with a further 17 patients (24.3%) receiving initial surgical care the day after admission. Amputation was required for 14 patients (20.0%) and 47 patients (67.1%) received vacuum-assisted wound therapy (VAC) to accelerate wound healing.

### 3.2. LRINEC Initial and in the Course of Time

The mean LRINEC of all patients at the time of hospital admission was 6.0 ± 2.9 (range 0–12). Twenty-six patients with histologically proven NF had an initial LRINEC of less than six (37.1%), of whom three patients (4.3%) died. Twenty patients were admitted with a LRINEC of six or seven (28.6%), and twenty-four patients (34.3%) had an initial LRINEC of equal or greater than eight. We can state that the LRINEC decreases in a highly significant manner, by 0.663 units on average, in the course of serial debridements (from hospital admission until the postoperative time after the fourth debridement) with a 95% confidence interval of 0.903 to 0.422 (*p* < 0.0005) (Figure 1 and Figure 2). The most significant changes are observed after the first debridement.

### 3.3. Factors Influencing the Course of the LRINEC

The change of the LRINEC (AD-0–AD-1) after the first debridement is particularly striking. We, therefore, searched for factors that could possibly exert an influence on this change; however, using linear regression analysis, no significant influence could be detected for either the time delay of the first debridement (*p* = 0.683) or patient age at hospital admission (*p* = 0.688). The influence of gender and the presence of diabetes mellitus or coronary heart disease on the initial change in LRINEC were also investigated by *t*-tests, but no significant influence on the initial change of LRINEC could be detected (*p* = 0.679, *p* = 0.908 and *p* = 0.561).

### 3.4. LRINEC as a Predictor of Lethality

A t-test revealed significant correlation between the initial LRINEC and the patient outcome. There were significantly higher LRINEC values for patients with exitus letalis than for those with positive disease progression (*p* = 0.006). Furthermore, we built receiver operating characteristic (ROC) curves to determine an appropriate cut-off value for the initial LRINEC score regarding lethality (Figure 3). The area under the ROC curve was 0.677 (*p*-value = 0.021). Thus, the area under the curve (AUC) was significantly greater than 0.5 and the initial LRINEC (AD-0) can be considered a significant predictor of lethality. In the context of our studies, a cut-off value of greater than 6.5 (i.e., 7, because LRINEC score is an integer) resulted in an optimal constellation of sensitivity (70%) and specificity (60%), with a positive predictive value of 41.2% and a negative predictive value of 83.3%.

To analyze the relationship between the initial LRINEC and the survival time at the established cut-off value of 7 LRINEC points, a Kaplan–Meier evaluation was performed, and the corresponding Kaplan–Meier curves were calculated (Figure 4). Using the Tarone–Ware test, a significant difference between the two Kaplan–Meier curves was calculated (*p* = 0.034).

## 4. Discussion

The diagnosis of NF poses great challenges to physicians, as the soft tissue infection first spreads to deeper layers, making the impression of the extent of the infection deceptive and the initial nonspecific symptoms easily confusing [16,17,18]. Especially in severely ill patients, it is of enormous importance to identify high-risk patients at an early stage and to be able to use an objective parameter to assess the severity of the disease in addition to the clinical assessment. For this purpose, the LRINEC score was launched by Wong et al. in 2004 to allow both risk stratification of patients and differentiation of NF from non-necrotizing soft tissue infections [14]. Wong et al. themselves viewed the LRINEC as a robust score that can even detect early cases of NF [14]. On the other hand, some studies describe the LRINEC as inadequate for distinguishing NF from other soft tissue infections, due to an insufficient sensitivity [19,20,21,22]. Furthermore, Harasawa et al. recently established another clinical score, to enhance the sensitivity, specificity, and especially the negative prognostic value, compared with the LRINEC score [23]. In the literature, data ranging from 43% to 80% are found for the sensitivity of the score [19,20,24]. To date, the highest sensitivity for the LRINEC was found in the original study by Wong et al. (2004), at 89.9% [14]. In the present study, LRINEC was less useful for diagnosing necrotizing fasciitis. Only 44 patients (62.9%) achieved a LRINEC equal or greater than six at initial presentation, although NF was histologically confirmed in all patients during follow-up. Accordingly, 26 patients (37.1%) had a score of less than six and would be incorrectly classified as low risk for the presence of NF according to the LRINEC scoring system. Three of these 26 patients died during the inpatient stay. One patient in the collective even had a LRINEC score of 0 points. This phenomenon is also described by Wilson et al. in their publication [25].

In this study, the LRINEC score was retrospectively calculated postoperatively after surgical debridement in order to demonstrate the progression of the score over time. It was shown that the score decreased significantly by an average of 0.663 LRINEC points during each debridement. Since the LRINEC is a measure of the severity of sepsis, it can be concluded that surgical therapy contributes significantly to lowering the LRINEC and, thus, to improving the general condition of the patient, which in turn underlines the importance of surgical therapy. Looking more closely at the progression of LRINEC, it can also be seen that the greatest change can be recorded during the initial debridement, suggesting that the initial intervention brings the greatest benefit to the patient. In addition to the effect of surgery, there are also other important aspects, such as the time from admission to surgery, the start of antibiotic therapy and/or further intensive care measures.

However, to lower the score sufficiently, and thus bring about patient recovery, the first debridement is not sufficient. Instead, regular evaluations and revisions, at intervals of 24–36 h, are necessary to completely remove the avital tissue [26].

In numerous studies, the lethality ranges from 20 to 30% and could reach up to 100%, if there are delays in diagnosis and treatment [2,3,4,5]. The lethality in the present study was 28.6%. Adequate therapy was given in a timely manner.

In a further step, we searched for patient characteristics or factors that exert an influence on LRINEC(-progression). Delayed initial debridement is associated with increased morbidity and lethality [26,27,28,29]. Based on this, our study aimed to examine whether the positive effect of expeditious debridement also positively influences changes in LRINEC. However, no significant relationship could be elicited in this regard. Demographic characteristics such as age and gender were also the subject of retrospective and prospective studies with regard to disease progression in patients with NF. Several authors reported that advanced age is an independent predictor of lethality in patients with NF [30,31,32]. Elliott et al. further concluded that, in addition, NF is associated with increased lethality in women [31]. However, in the present study, neither age nor patient sex significantly influenced the change in the LRINEC score (AD-0–AD-1). Furthermore, comorbidities, especially diabetes mellitus and coronary artery disease, are often discussed as risk factors for NF, which according to several studies, should exert a negative influence on the patient outcome [33,34,35]. However, neither diabetes mellitus, nor coronary heart disease could be shown to influence the change in LRINEC score (AD-0–AD-1) in this study. In a further step, the prognostic significance of the LRINEC in terms of patient outcome was investigated. In view of the fact that NF is a disease process with a lethality rate that cannot be neglected, an elevated LRINEC score in critically ill patients was already identified as a prognostically unfavorable marker in numerous publications [2,33,36]. El-Menyar et al. also described a significantly higher lethality in patients with a LRINEC equal or greater than six in their patient population [3]. Here, a LRINEC cut-off value of eight points was set for predicting hospital lethality, with a sensitivity of 81%, and a specificity of only 36% [3]. Our study also demonstrated a statistical correlation between initial LRINEC and lethal disease progression. To predict lethality, a cut-off value was calculated of 6.5 points (≥7 LRINEC points) with a sensitivity of 70%, specificity of 60%, positive predictive value of 41.2%, and negative predictive value of 83.3%.

The present study design has some limitations, especially due to its retrospective character. Since the data collection was based on patient records that were not always seamless, a complete recording of all relevant parameters was not possible in all cases, which is why retrospective analyses only allow statements of a low level of evidence. Especially when it comes to the calculation of the LRINEC score, laboratory results could not be collected from each patient at exactly the same point in time. Furthermore, the present study is based on a relatively small case series, which means that all relevant results of this study can only be generalized to a limited extent. Another weakness arises from the heterogeneous patient population with its different clinical courses and the decreasing sample size due to lethality and exclusion in the course of time.

To summarize the study, we present the LRINEC over the course of time. The LRINEC significantly decreases after any surgical debridement. However, surgical interventions can also lead to an increase in the LRINEC score, for example due to a drop in hemoglobin or a slight increase in CRP. A relevant drop in hemoglobin is usually compensated for by administering erythrocyte concentrates. In addition to the initial CRP values, only postoperative CRP values are compared, so that a possible postoperative CRP increase can be neglected in this case of the evaluation. Nevertheless, this must be taken into account when considering the LRINEC score. Overall, the surgical treatment of NF is of enormous importance.

Furthermore, this study was able to illustrate that the lethality of necrotizing fasciitis is still very high, despite adequate and early therapy; the medical team faces great challenges in the treatment of these fulminant soft tissue infections. In terms of diagnostic efficacy, the LRINEC proved to be unreliable in our work due to lack of sensitivity, a high rate of false–negative results, and a low positive predictive value. Accordingly, LRINEC should only be used as an adjuvant and with simultaneous consideration of all clinical parameters. Furthermore, as previously shown in other studies, we identified the initial LRINEC value at diagnosis as an independent prognostic marker, the level of which correlated significantly with patient outcome. To predict lethality, we set a cut-off value of seven LRINEC points in this regard. For treating physicians, this means that, in conjunction with a corresponding clinical expression, patients with a high LRINEC (≥7) must be immediately referred to adequate care centers having intensive care units and a plastic surgery department. Surgical debridement must be initiated as soon as possible to prevent a lethal course. Furthermore, the study showed that the course of LRINEC is not influenced by any of the investigated parameters and no conclusions on the clinical course can be drawn based on the LRINEC change. However, before definitive statements can be made regarding the suitability of LRINEC as a progression parameter and to substantiate the present data, the score needs to be prospectively tested in a large collective.

## 5. Conclusions

The LRINEC significantly decreased after surgical debridement in this study.

An initial LRINEC equal or greater than seven is an independent prognostic marker for lethality and can help to identify high-risk patients.

## Figures and Tables

**Figure 1 jcm-11-03583-f001:**
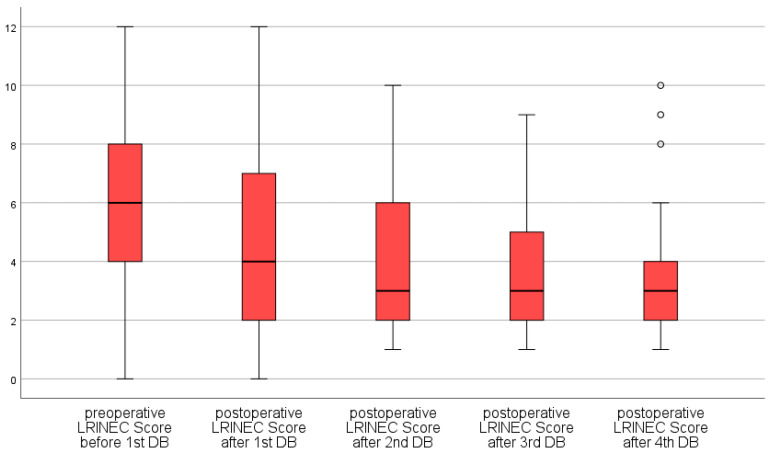
LRINEC in the course of time as a boxplot diagram.

**Figure 2 jcm-11-03583-f002:**
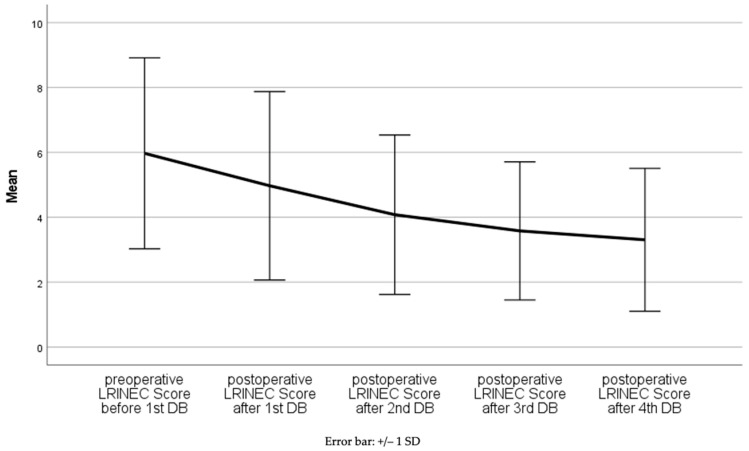
Mean LRINEC in the course of time.

**Figure 3 jcm-11-03583-f003:**
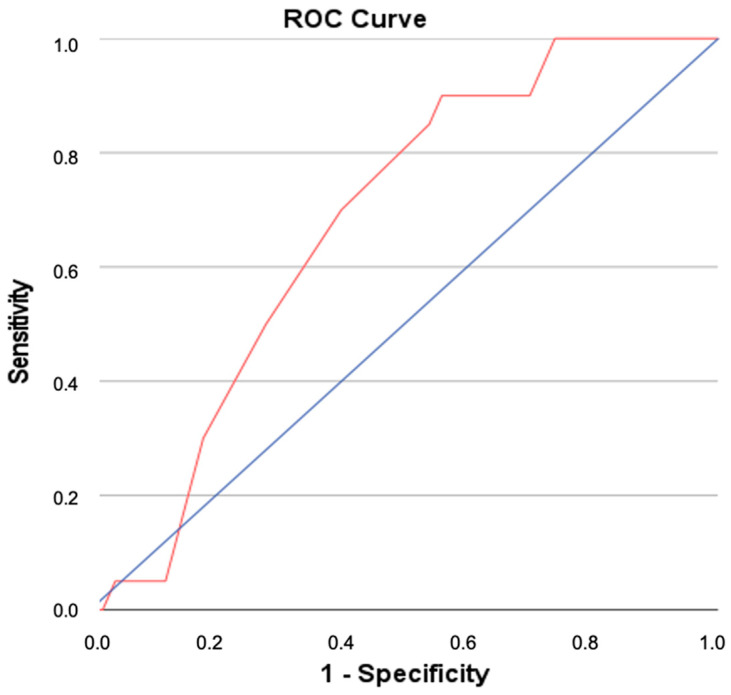
Receiver–operator curve for LRINEC for predicting (Area under the curve was 0.677 (*p*-value = 0.021)) lethality.

**Figure 4 jcm-11-03583-f004:**
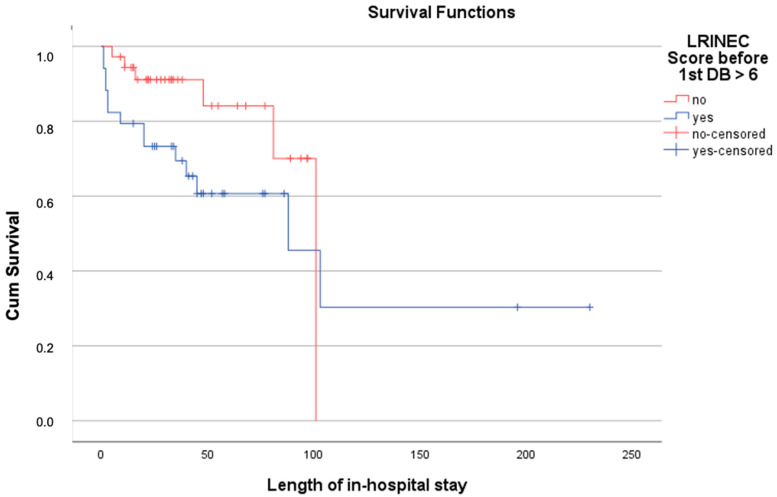
Survival analysis for hospital survival of patients with NF.

**Table 1 jcm-11-03583-t001:** The laboratory risk indicator for necrotizing fasciitis score (based on [14]).

Variable	Value	Score
C-Reactive protein (mg/L)	≤150	0
	>150	4
Total white blood cell count (1000 cells/µL)	<15	0
	15–25	1
	>25	2
Hemoglobin (g/dL)	>13.5	0
	11–13.5	1
	<11	2
Sodium (mmol/L)	≥135	0
	<135	2
Creatinine (mg/dL)	≤1.6	0
	>1.6	2
Glucose (mg/dL)	≤180	0
	>180	1

**Table 2 jcm-11-03583-t002:** LRINEC risk assessment (based on [14]).

Risk Category	LRINEC Points	Probability for Presence of NF
Low	≤5	<50%
Medium	6–7	50–75%
High	≥8	>75%

**Table 3 jcm-11-03583-t003:** Recording times for the calculation of the LRINEC score and its related sample sizes.

Description	Laboratory Values Are from the Following Period:	Sample Size
**Acquisition date 0** (AD-0)	within 48 h before 1st DB (LRINEC initial)	70 patients
**Acquisition date 1** (AD-1)	1st or 2nd postoperative day after 1st DB	70 patients
**Acquisition date 2** (AD-2)	1st or 2nd postoperative day after 2nd DB	63 patients
**Acquisition date 3** (AD-3)	1st or 2nd postoperative day after 3rd DB	50 patients

## Data Availability

Not applicable.

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
