# Peer review of "The LRINEC Score—An Indicator for the Course and Prognosis of Necrotizing Fasciitis?"

_jcm, 2022, doi:10.3390/jcm11133583_

Round 1
Reviewer 1 Report
Dear authors:
In my opinion yor paper show an interesting point of view about "LRINEC" as a tool for evaluating evolution of the patients affected with NF.
Only a comment. In yor "Discussion", line 249, you say that "suggesting that the initial intervention brings the greatest benefit to the patient". I partially agree with your finding, but you should keep in mind that so many factors such as time from admission to surgery, type of antibiotic therapy and other factors (HBO, VAC...) may jeopardize this finding. But you finally recognize some limitations (pg 9 number 284-293) and I am confortable with it.
Author Response
Dear Reviewer,
thank you very much for your comment.
In our study, the greatest change of the LRINEC-Score occurred after the first debridement. In addition to the effect of surgery, there are also other important aspects like the time from admission to surgery, the begin of antibiotic therapy or further intensive care measures.
We adapted the manuscript.
Kind Regards
Reviewer 2 Report
The present clinical retrospective study focuses on the LRINEC score in diagnosing and monitoring necrotising fasciitis (NF). NF is still a very rare, but still very serious desesase with a high legality and a challenging clinical therapy. The LRINEC is one of the few clinical scores in diagnosing NF. However, since its publication in 2004, there is an ongoing debate about the usefulness because of the lack of both sensitivity and specificity regarding the diagnostic power for NF. The present study is the first study evaluating the LRINEC score over time during the treatment period.
Despite the low evidence in the literature for NF and therefore a tremendous need for more studies, the present study has several severe limitations:
1.) Material & Methods: NF is a clinical diagnosis. However, the authors state that "Diagnosis was confirmed by histologic examination" (l. 95). As the histologic appearance of NF might be the same as e.g. the streptococcal toxic shock syndrome (STSS), I would like the authors to provide information about the exact definition of Necrotizing Fasciitis in their study.
2.) Results: The authors describe a time-dependent decrease of the LRINEC score over time/amount of surgeries in their collective. However, over the time points from first debridement to third debridement, the study collective decreases from 70 patients to 50 patients. I could not find any explanation in the study for this decrease. Possible explanations are both death and the "cure" of NF and therefore no need of further debridements. Are there any other reasons? As the total lethality rate was 20 patients, I have to assume that all patients died in between the first to third debridement? Wasn't there any death afterwards, but still prior to discharge? The authors state that "laboratory results could not be collected from each patient at exactly the same point of time" (ll 288-289), but do not provide detailed information about this bias.
3.) Study design: The authors linked the time points of LRINEC assessment to the time point of surgical debridement. This is one of the major weaknesses of the study. I would prefer rather a regular time-dependent follow up, e.g. day 2, day 4, day 6.. after admission to the hospital. The linking to the surgery provides a significant bias, as the interval in between the surgeries differs in the patient cohort and therefore harms the comparison of the data.
4.) Conclusion: As there is no control group with e.g. antibiotics only (without surgical intervention), the authors can - in my point of view - not conclude, that "the LRINEC significantly decreases after any surgical debridement, which underscores the tremendous importance of prompt, aggressive surgical debridement in the treatment of necrotizing fasciitis" (ll 296-299). In contrast, a surgical interventions themselves often increase inflammatory parameters as the CRP, they might rather increase the LRINEC score temporarily. Beside that, I do fully agree about the importance of surgical debridement in treating NF.
5.) Literature: All cited literature seems to be relevant for the paper. However, there is a recent meta analysis evaluating the LRINEC score in diagnosing NF, which is not discussed at all in this study and which describes a poor sensitivity of the LRINEC score (Fernando SM, Tran A, Cheng W, Rochwerg B, Kyeremanteng K, Seely AJE, Inaba K, Perry JJ. Necrotizing Soft Tissue Infection: Diagnostic Accuracy of Physical Examination, Imaging, and LRINEC Score: A Systematic Review and Meta-Analysis. Ann Surg. 2019 Jan;269(1):58-65. doi: 10.1097/SLA.0000000000002774. PMID: 29672405). Furthermore, other clinical scores than the LRINEC score have been established recently and have to be included to the discussion, as they might enhance the sensitivity, specificity and especially the negative prognostic value compared to the LRINEC score (Harasawa T, Kawai-Kowase K, Tamura J, Nakamura M. Accurate and quick predictor of necrotizing soft tissue infection: Usefulness of the LRINEC score and NSTI assessment score. J Infect Chemother. 2020 Apr;26(4):331-334. doi: 10.1016/j.jiac.2019.10.007. Epub 2019 Nov 8. PMID: 31711831.). Minor spelling mistake: l. 232 Wong instead of Wilson (reference 14).
To summarize, the LRINEC score seems not to be precise for the evaluation of NF and the present study does not provide sufficient and relevant new data. The decrease of the LRINEC score over time/surgical debridements in the surviving cohort seems to be logical and also interesting, but without high impact for the clinical practice.
Author Response
Dear Reviewer,
thank you for your comments and suggestions.
I integrated them in the manuscript.
Please see the attachment.
Kind Regards

Reviewer 3 Report
It is well written paper but conclusion can be shortened in 3 lines and summary can be part of discussion
Author Response
Dear Reviewer,
thank your very much for your comments and suggestions.
According to your review, the conclusion was shortened and the summary was integrated in the discussion. Kind RegardsRound 2
Reviewer 2 Report
Thank you four reply. I would still suggest to modify 2 parts:
1.) I would still recommend to re-phrase the sentence "The LRINEC significantly decreases after surgical debridement, which underscores 349 the tremendous importance of prompt and adequately aggressive surgical debride- 350 ment of patients with NF." (ll. 349-351). As there is no control group due to the study design, one can not conclude a causal relationship between the time point of surgery and the LRINEC score, as the score might also decrease in patients without re-surgery.
2.) I would still like the manuscript to present more details about the decrease of the study collective. As the authors state in their reply, the "study collective decreases due to death but mainly due to drop out due to the lack of data". I would recommend to provide exact numbers of drop-outs either because of death or because of missing data.
Author Response
Dear Reviewer,
Thank you again for your comments and suggestions.
Please see the adaptions in the file attached.
Best regards
